# The Role of Tumor Microenvironment in Pancreatic Cancer Immunotherapy: Current Status and Future Perspectives

**DOI:** 10.3390/ijms25179555

**Published:** 2024-09-03

**Authors:** Fotini Poyia, Christiana M. Neophytou, Maria-Ioanna Christodoulou, Panagiotis Papageorgis

**Affiliations:** 1Tumor Microenvironment, Metastasis and Experimental Therapeutics Laboratory, Basic and Translational Cancer Research Center, Department of Life Sciences, European University Cyprus, Nicosia 2404, Cyprus; f.poyia@external.euc.ac.cy; 2Apoptosis and Cancer Chemoresistance Laboratory, Basic and Translational Cancer Research Center, Department of Life Sciences, European University Cyprus, Nicosia 2404, Cyprus; c.neophytou@euc.ac.cy; 3Tumor Immunology and Biomarkers Laboratory, Basic and Translational Cancer Research Center, Department of Life Sciences, European University Cyprus, Nicosia 2404, Cyprus; mar.christodoulou@euc.ac.cy

**Keywords:** pancreatic cancer, immunotherapy, tumor microenvironment

## Abstract

Pancreatic cancer comprises different subtypes, where most cases include ductal adenocarcinoma (PDAC). It is one of the deadliest tumor types, with a poor prognosis. In the majority of patients, the disease has already spread by the time of diagnosis, making full recovery unlikely and increasing mortality risk. Despite developments in its detection and management, including chemotherapy, radiotherapy, and targeted therapies as well as advances in immunotherapy, only in about 13% of PDAC patients does the overall survival exceed 5 years. This may be attributed, at least in part, to the highly desmoplastic tumor microenvironment (TME) that acts as a barrier limiting perfusion, drug delivery, and immune cell infiltration and contributes to the establishment of immunologically ‘cold’ conditions. Therefore, there is an urgent need to unravel the complexity of the TME that promotes PDAC progression and decipher the mechanisms of pancreatic tumors’ resistance to immunotherapy. In this review, we provide an overview of the major cellular and non-cellular components of PDAC TME, as well as their biological interplays. We also discuss the current state of PDAC therapeutic treatments and focus on ongoing and future immunotherapy efforts and multimodal treatments aiming at remodeling the TME to improve therapeutic efficacy.

## 1. Introduction

Pancreatic cancer (PC) is the fourth leading cause of cancer-associated deaths worldwide both in men and women [1]. The most common form of PC is pancreatic ductal adenocarcinoma (PDAC), accounting for 90% of all cases [2]. With persistently escalating incidence and minimal change in mortality rates, PDAC is predicted to become the second most frequent cause of cancer death within the next six years [3]. The average lifetime risk of developing PC is around 1.5%, which translates to 1 in 64 people. The aggressive behavior and the rapid development of metastases explain why PC patients have a 5-year overall survival (OS) rate of less than 13% [4,5]. The progression from stage I to stage IV is estimated to last just over a year. Moreover, PDAC symptoms, such as type-2 diabetes outbreak, abdominal and back discomfort, lack of appetite, and weight loss, are not specific and are often misinterpreted, leading to poor and late diagnosis. The risk of developing PC increases with age, with the average age at the time of diagnosis to be 70 years [6,7]. Furthermore, only 10% of PC incidents are hereditary, and approximately 90% are sporadic. Among patients with PC, 90% carry a *KRAS* mutation, which is considered a driver gene for PC progression, and 50–80% have inactivating mutations in *TP53*, *CDKN2A*, and *SMAD4* [8]. While novel treatments have significantly improved the OS rate in other cancers, PC still constitutes one of the deadliest forms of malignancy, with a median survival of 15.5 months after surgery. Only up to 10% of patients who receive a timely diagnosis become disease-free after treatment. Consistent with this, patients who are diagnosed before metastatic growth have an average survival time of just 3 to 3.5 years, whereas patients diagnosed with final stage of PC have a life expectancy of about 3–5 months [2,9]. Surgical resection is the mainstay of curative treatment for patients with localized pancreatic tumors. In cases of borderline resectable PC (BRPC) or unresectable locally advanced PC (LAPC), pre-operative neoadjuvant therapy using FOLFIRINOX (5-fluorouracil, leucovorin, oxaliplatin, and irinotecan) or gemcitabine/nab-paclitaxel), with or without radiation therapy, is used to downsize the tumor to facilitate surgical resection. These are also administered as the standard-of-care treatment for patients with metastatic PDAC, whereas pembrolizumab is considered for <1% of metastatic patients with high microsatellite instability (MSI-H) or mismatch repair deficiency (dMMR). The poly (ADP-ribose) polymerase (PARP) inhibitor Olaparib is used as maintenance therapy in patients with BRCA1/2 mutations after initial chemotherapy response [10].

This review summarizes the current state of PC treatment, the main characteristics of pancreatic cancer TME, and their implications in exploring new potential therapeutic targets and combination approaches. Specifically, we focus on immunotherapy and the roles of PC TME in hindering its efficacy and discuss the latest pre-clinical and clinical evidence regarding multimodal treatment strategies to improve immunotherapy outcomes in PC patients. 

## 2. Current Therapies for Pancreatic Cancer

Different types of treatments are available for people with PC, with the only curative approach being surgical management. Importantly, if PC has spread, palliative treatment can only improve patients’ quality of life by controlling the symptoms of this disease. In this section, we discuss the current types of standard-of-care treatment for PC.

### 2.1. Surgery

While surgery is the only available treatment with curative potential, 85% of newly diagnosed pancreatic tumors are considered unresectable due to late diagnosis and locally advanced disease or metastasis to distal organs [11,12,13]. However, up to 80% of the patients that are eligible for surgery may relapse and die after the operation [9,14]. In cases where cancer has spread throughout the pancreas but is still resectable, a total pancreatectomy is performed, and post-operative patients receive pancreatic enzymes for life [12]. Therefore, even though surgical management plays a vital role in PC cure, few patients can benefit long-term due to compromised quality of life and other surgery-related risks, including post-operative infections [10]. Hence, administration of chemotherapy after surgical management of PDAC has been shown to lower the risk of recurrence and improve survival rates. 

### 2.2. Chemotherapy

Chemotherapy has been typically the standard-of-care treatment for primary and metastatic PDAC. Gemcitabine is the reference treatment as anti-cancer chemotherapy in PC patients that are not eligible for combination chemotherapy [15]. Additional treatment protocols, including neoadjuvant chemotherapy and adjuvant chemotherapy for resected PC, are often combined with surgery to increase the rate of successful resection and extent of survival. Unfortunately, even with cytotoxic chemotherapy, the OS of locally advanced PC patients with metastatic disease is only 5–6 months, with a response rate of 5.4% [16,17]. NALIRIFOX, an irinotecan liposome (ONIVYDE) with oxaliplatin, fluorouracil, and leucovorin, was approved by the FDA (Food and Drug Administration) as a first-line treatment for metastatic pancreatic adenocarcinoma. The NAPOLI 3 clinical trial (NCT04083235) demonstrated a statistically significant improvement in overall survival for the NALIRIFOX arm over the Nab-paclitaxel plus gemcitabine arm (11.1 months vs. 9.2 months). The application of adjuvant chemotherapy was supported by the phase III CONKO-001 randomized trial (ISRCTN34802808), which showed a significant benefit of adjuvant gemcitabine after PC resection against surgery alone. A prolonged disease-free survival (DFS) (13.4 vs. 6.7 months) and 5-year OS (20.7% vs. 10.4%) was observed, including patients with R0 or R1 resected tumors. Long-term follow-up also displayed an increased 10-year OS of 5%. Furthermore, the ESPAC randomized trials aimed to identify the most effective chemotherapy scheme. The most recent ESPAC-4 trial (ISRCTN96397434) of Neoptolemos et al. underlined the advantage of gemcitabine plus capecitabine in the adjuvant setting after surgery, resulting in median survival of 26 months and 5-year survival of 30% [18]. 

Progress has also been achieved with FOLFIRINOX based on the PRODIGE-24/CCTG clinical trial (NCT01526135), which compared the outcomes of FOLFIRINOX against gemcitabine in patients with resected PDAC. The results indicated a clear improvement in OS using FOLFIRINOX against the gemcitabine group (54.4 vs. 35 months). The 5-year disease-free survival rate was 26.1% for FOLFIRINOX-treated patients and 19.0% for gemcitabine [19]. However, FOLFIRINOX administration was correlated with an increased risk of complications. Improved survival was observed in metastatic patients after combination treatment of gemcitabine and nab-paclitaxel, with fewer side effects than FOLFIRINOX [20].

Although adjuvant treatment may provide a survival benefit, about 74% of patients still relapse within two years [21]. A meta-analysis of 13 trials demonstrated downstaging of unresectable tumors after neoadjuvant FOLFIRINOX treatment, achieving an R0 resection rate of 40% [11]. However, resectable disease may become unresectable upon receiving neoadjuvant therapy due to complications which can prevent surgical management [12]. Further clinical studies are needed to reach optimal treatment protocols for administering chemotherapy in the neoadjuvant setting.

Up to 80% of patients do not respond to neoadjuvant chemotherapy to become eligible for surgery. The standard treatment of these tumors using systemic chemotherapy, commonly FOLFIRINOX, gemcitabine, and/or nab-paclitaxel, aims to control the disease which has already metastasized [22]. However, the OS of locally advanced PDAC (LAPC) patients remains below one year [23]. Interestingly, Shelemey et al. reported a case of shrinkage of an adenocarcinoma mass on the pancreatic tail and of liver metastases after FOLFIRINOX administration. Importantly, upon completion of 37 cycles of FOLFIRI (FOLFIRINOX without oxaliplatin), the pancreatic mass disappeared, the liver metastasis decreased, and no recurrence was observed [24]. 

### 2.3. Radiotherapy

External radiation therapy may be a treatment option for PC patients depending on the stage of the disease; specific tumor characteristics, such as size and location; and the patient’s overall health. Although the aim of radiation is to control cancer development and relieve patient symptoms, its use has been limited due to the inability to administer effective radiation doses in the pancreatic tumor. The inherent resistance of the pancreatic tumor, the intraperitoneal tumor location, and the neighboring organs impose barriers on the effective and targeted application of radiation therapy [25]. To address these problems, therapies have been developed to decrease tumor volume using the FDA-approved stereotactic body radiotherapy (SBRT). It has been shown that SBRT is beneficial as an adjuvant treatment in high-risk PDAC patients with affected tumor margins after surgery [26]. Additionally, chemoradiotherapy, the use of radiotherapy as neoadjuvant treatment together with chemotherapy, has been assessed in patients with borderline resectable PC. The combination of chemotherapy with photon radiotherapy has demonstrated improved local control rates, but no OS benefit if no surgical resection is followed [27]. A phase I/II study (NCT00438256) by Hong et al. showed that preoperative chemoradiation along with proton radiotherapy and capecitabine followed by early surgery is feasible with low toxicity levels [28,29]. Several clinical trials are currently underway to evaluate long-term survival benefits from the combination of radiotherapy with other anti-cancer treatments. 

### 2.4. Targeted Therapies

Developments in whole-genome sequencing approaches have aided the design of personalized targeted therapies through mapping of key genetic alterations that drive PC progression. Mutations that are produced by genomic instability frequently produce cancer cell vulnerabilities that could be key for effective anti-cancer therapies. Diverse targeted agents have been assessed either alone or in combination with chemotherapeutic drugs against PDAC [30]. Unfortunately, most of these approaches have failed to prolong patient survival, mainly due to the hypovascular and desmoplastic nature of PC’s impenetrable stroma, as is described in more detail below [31]. A series of phase III clinical trials (NCT00088894, NCT01214720, NCT00471146, NCT00541021) in patients with advanced or metastatic PC failed to enhance OS after treatment with VEGF inhibitors, including bevacizumab (humanized anti-VEGF-A monoclonal antibody), sorafenib (VEGF-R inhibitor), and axitinib (tyrosine kinase inhibitor, TKI), along with gemcitabine or gemcitabine/erlotinib [32,33,34,35,36]. Furthermore, therapies targeting key signaling pathways in PDAC, including the anti-insulin-like growth factor 1 (IGF-1) receptor using the antibodies ganitumab and cixutumumab, the multi-kinase inhibitor masitinib, and the phosphoinositide 3-kinase (PI_3_K) inhibitor rigosertib, have been proven ineffective in randomized clinical trials [30]. Erlotinib is the only targeted agent that has exhibited a statistically significant, yet clinically modest, effect in patient survival. In a randomized trial (NCIC CTG PA.3) by Moore et al., it was reported that a combination of erlotinib with gemcitabine rendered a survival benefit of 2 weeks compared to gemcitabine alone [37]. 

## 3. Pancreatic Tumor Microenvironment and Opportunities for Therapeutic Interventions

Understanding the complex molecular composition and cellular interactions between cancer cells and the TME is of paramount importance for improving outcomes of existing therapeutic approaches and for designing novel personalized and precision therapeutic strategies for PC patients. Here, we describe the major components of the PDAC microenvironment, analyze the interactions of malignant cells with stromal and immune cells during PC formation and progression, and discuss their therapeutic implications.

### 3.1. Non-Cellular Components and Desmoplasia in the Pancreatic TME 

Throughout all stages of PDAC growth, tumor cells are not only in physical, but also in biological contact with the stroma via secreted factors mediating cell-to-cell communication. These continuous interactions affect the TME during oncogenesis and the accompanying stromagenesis [38,39,40]. Tumor desmoplasia is a phenomenon which refers to the growth of dense connective tissue or stroma around a tumor mass. It is characterized by the proliferation of fibroblasts and the production of extracellular matrix (ECM) components, such as collagen, resulting in a fibrous or hard tissue environment around the tumor, which is a hallmark of primary and metastatic PC [41]. ECM is a high-density network made up mainly of matrix proteins, including I, III, and IV collagens, which are secreted by cellular components of the TME, such as fibroblasts and pancreatic stellate cells (PSCs), along with hyaluronic acid, fibronectin, and glycosaminoglycan, and can collectively represent up to 90% of the tumor mass [13]. Specifically, type IV collagen has been proposed as a potential serum biomarker in predicting PC patient survival following a surgical operation [42]. Excessive deposition of ECM components in the TME results in increased tumor stiffness, solid stress, and interstitial fluid pressure (IFP), as well as application of mechanical forces by the surrounding stroma which compress blood vessels, leading to hypoperfusion, hypoxia, and decreased infiltration of cytotoxic immune cells [43,44]. Importantly, this highly dense ECM tissue also behaves as a physical barrier diminishing the effective penetration of anti-cancer drugs [45,46,47,48]. Losartan, an angiotensin II receptor blocker which suppresses TGF-β activity in PC, was shown to reduce the levels of collagen and hyaluronan in PDAC models. As a result, losartan improved vascular perfusion and, thus, the delivery and efficacy of cytotoxic agents, such as 5-FU and Doxorubicin. Interestingly, tumors were significantly smaller in mice administrated with losartan combined with either 5-FU or Doxorubicin instead of pancreatic tumors treated with one of them as a monotherapy [49,50]. Furthermore, Anup et al. indicated that an analysis of metastatic PDAC patients treated with FOLFIRINOX plus losartan revealed a longer progression-free survival (PFS) than the control group, although no statistical significance was observed [51]. Overall, it is unambiguously accepted that pancreatic tumor stiffness and desmoplasia play crucial roles in promoting disease aggressiveness, therapeutic resistance, and poor prognosis of PC patients [45].

### 3.2. Non-Immune Cellular Components in the TME

The excessive production of ECM components resulting in the formation of the extremely dense PC stroma is mainly mediated by PSCs which represent the main fibroblastic cell type in PDAC, along with cancer-associated fibroblasts (CAFs). CAFs are a heterogeneous population of fibroblasts found in the pancreatic tumor stroma. They can originate from various sources, including resident fibroblasts, mesenchymal stem cells, and even PSCs that have been activated and further modified in the tumor microenvironment. They can be further subcategorized in myofibroblastic CAFs (myCAFs) and inflammatory CAFs (iCAFs) [52,53]. MyCAFs highly express α smooth muscle actin (α-SMA) and are often localized close to PC cell clusters, whereas inflammatory CAFs (iCAFs) are located more distantly from the tumor cells in the desmoplastic stroma [54].

#### 3.2.1. Pancreatic Stellate Cells (PSCs)

In the healthy pancreas, PSCs remain in a quiescent state. Upon pancreatic injury, inflammation, or tumor formation, PSCs become activated, transforming into a myofibroblast-like phenotype, which is associated with increased production of ECM components. Activated PSCs within and surrounding the tumor produce collagen and other subcomponents of the ECM, which contribute to desmoplasia, increased solid stress, and poor vascularity due to vessel compression, a characteristic feature of PDAC [55]. It has also been suggested that PSCs are involved in cancer initiation, angiogenesis, epithelial-to-mesenchymal transition (EMT), local invasion, and metastasis of PC cells [56] by expressing paracrine molecules, such as transforming growth factor-β (TGF-β) and platelet-derived growth factors (PDGFs) [57,58]. Moreover, PSCs have been shown to support PC progression by increasing the number of immunosuppressive cells and inhibiting infiltration of cytotoxic CD8^+^ T cells [54,59,60]. 

#### 3.2.2. Myofibroblastic and Inflammatory CAFs

Based on initial evidence that the desmoplastic stroma acts as a physical barrier to compromise efficient cytotoxic drug delivery in pancreatic tumors [61,62], subsequent efforts were aimed to eliminate or target myofibroblastic cancer-associated fibroblasts (myCAFs) from pancreatic cancers. However, these studies suggested that local depletion or inhibition of α-SMA+ myCAFs in a murine PDAC model reduced desmoplasia, but were associated with increased tumor aggressiveness, immunosuppression, and shorter survival instead of promoting anti-tumor effects [63,64]. Collectively, these studies have shown that myCAFs could restrain tumor growth and that therapeutic strategies targeting them may only be considered in the context of combination therapies.

Inflammatory CAFs (iCAFs) also originate from activated PSCs and other fibroblasts within the tumor microenvironment, but are distinguished by their inflammatory profile based on the expression of inflammatory cytokines and chemokines. For example, secretion of IL-6 by iCAFs acts synergistically with IL-10 and TGF-β to inhibit dendritic cell proliferation, therefore inhibiting tumor-antigen presentation [65,66,67]. They may also express fibroblast-specific markers like fibroblast activation protein (FAP) and PDGFRβ, but are primarily defined by their secretion profile. By promoting inflammation, they influence the recruitment and activation of M2-type tumor-associated macrophages (TAMs), myeloid-derived suppressor cells (MDSCs), and regulatory T cells (Tregs), which can contribute to immunosuppression and tumor immune evasion [54,68]. 

#### 3.2.3. Endothelial Cells

Moreover, endothelial cells play crucial roles in the development and structure of blood vessels in pancreatic tumors [69]. Studies have shown that angiogenesis in PDAC demonstrates abundant production of the vascular endothelial growth factor (VEGF) by endothelial as well as PC cells under the control of hypoxia-inducible factor 1 subunit alpha (HIF1α) and signal transducer and activator of transcription 3 (STAT3) under hypoxia conditions [70,71,72]. Unfortunately, so far, no anti-angiogenic therapy has been clinically effective in PDAC.

### 3.3. Immune Cell Populations in the PDAC TME

It becomes increasingly evident that the functional network of interactions between tumor, stromal, and immune cells supports the progression of PC. During pancreatic tumorigenesis, the immune system may act as a double-edged sword; certain immune components can suppress tumor growth or progression by recognizing and eliminating mutated cells, while others can promote an immunosuppressive and pro-tumorigenic environment. Although immune cell populations account for up to 50% of the total cell number in PDAC, only a small subset are tumoricidal cells [73]. The major immune cell types in the PDAC TME that play crucial roles in these processes include macrophages, myeloid-derived suppressor cells (MDSCs), natural killer cells (NKs), neutrophils, dendritic cells (DC), and T lymphocytes.

#### 3.3.1. Tumor-Associated Macrophages (TAMs)

Tumor-associated macrophages (TAMs) are one of the most abundant immune cell types in the pancreatic TME; they originate from circulating monocytes and are recruited to the pancreatic tumor site by various chemokines and growth factors secreted locally [74,75,76]. They are considered a heterogenous population due to their plasticity and ability to switch between the anti-tumoral M1 and pro-tumoral M2 phenotypes depending on the conditions in the TME and activation signals [77,78]. The majority of TAMs in PDAC display the M2-polarized phenotype, characterized by the surface markers CD163 and CD206, and secrete IL-10 and TGF-β [79]. Importantly, Ino et al. reported that the tumor-infiltrating % of M1^high^/M2^low^ may act as an independent prognostic factor for OS in PDAC patients [80]. Macrophage depletion was found to reduce liver and lung metastasis in an orthotopic PDAC mouse model [81]. Several findings support that TAMs could also regulate PDAC metastasis through secretion of exosomes containing miRNAs that promote tumor cell migration, EMT, and ECM remodeling [82,83]. Finally, in an in vivo PDAC mouse model study, cytidine deaminase (CDA), a key metabolizer of gemcitabine, was found to be upregulated by TAMs, resulting in an anti-inflammatory macrophage phenotype and chemotherapy resistance [84].

#### 3.3.2. Myeloid-Derived Suppressor Cells

Myeloid-derived suppressor cells represent a mixture of immature myeloid cells with a critical role in immunosuppression in PC. They are abundantly found in PDAC and are dispersed throughout the tumor. Their accumulation is associated with the stage of the disease [73,85]. High levels of granulocyte macrophage colony-stimulating factor (GM-CSF) produced by tumor cells are associated with MDSC development and migration through the bloodstream [86]. Pylayeva et al. demonstrated that the oncogenic *KRAS^G12D^* mutation, present in more than 90% of PC cases, is responsible for the upregulation of GM-CSF [87]. It is also known that MDSC differentiation is triggered by the STAT3 signaling pathway upon IL-6 release from activated PSCs [88]. Furthermore, MDSCs can suppress CD4^+^ and CD8^+^ T cell responses via several mechanisms, first by upregulating PD-L1 and inhibiting T cell activation and tumor tolerance [89]. Moreover, MDSCs were shown to stimulate expansion of immunosuppressive Tregs [90] which, in turn, induce MDSCs to release reactive oxygen species (ROS), causing oxidative stress in T cells to further inhibit antigen-presenting proliferation [91]. In addition, MDSCs downregulate L-selectin in CD4^+^ and CD8^+^ T cells impairing T cell homing to lymph nodes [92]. 

#### 3.3.3. Natural Killer Cells

Natural killer (NK) cells account for 5–20% of human peripheral blood mononuclear cells (PBMCs). They are characterized by the expression of the natural cytotoxicity receptor (NCR) NKp46 and the neural cell adhesion molecule (NCAM/CD56) [93,94]. Upon activation, NKs secrete IFN-γ, GM-CSF, tumor necrosis factor-α (TNF-α), and chemokines that regulate the functions of other innate and adaptive immune cells [95]. The number of circulating NKs in PDAC is positively correlated with median patient survival [96]. Increasing evidence suggests that interactions within the pancreatic TME can regulate the phenotype and function of NKs. It was recently proposed that tumor cell-derived extracellular vesicles from the TME can functionally change NK cells by inhibiting the recognition and killing of cancer cells [97]. The activity of NKs was shown to be reduced in PDAC compared to peripheral blood leukocytes of healthy blood donors based on the production of lower levels of granzyme B and perforin, which are crucial for the elimination of cancer cells [98]. Human PC cells express Fas ligand, leading to apoptosis of tumor-infiltrating lymphocytes, including NKs [99]. Moreover, NKs’ recognition and killing abilities are impaired by IL-10, TGF-β, Indoleamine 2,3-dioxygenase (IDO) and metalloproteinases produced by PDAC cells [100]. Reduced levels of the activating receptor NKp46 are correlated with PC progression [101]. Finally, C-X-C motif chemokine receptor 2 (CXCR2) has been found to be essential for the recruitment of NKs into the TME. This chemokine receptor is downregulated in PDAC patients, resulting in limited NK cell infiltration [13]. 

#### 3.3.4. Neutrophils

Neutrophils, an essential component of the innate immune system, have evident anticancer activity and can induce phagocytosis as well as direct cytotoxic elimination of malignant cells. They infiltrate the TME upon interaction of CXCR2⁺ neutrophils with CXCL1/2 ligands [102]. During early stages of cancer development, tumor-associated neutrophils (TANs) can be distinguished according to their cytokine status, activation, and effects on cancer cells. N1 TANs regulated by IFN-α can exert anti-tumor effects. They exert cytotoxic action against tumor cells and prevent immunosuppression within the TME mainly by recruiting and activating CD8⁺ T cells [13,103]. On the contrary, N2 TANs induced by TGF-β undergo a phenotypic switch to a pro-tumoral phenotype, promoting tumor progression by remodeling the TME, whereas TGF-β blockade was able to reverse this effect in colorectal cancer in an in vitro study [90,104]. In addition, neutrophils contribute to tumor invasion and metastasis through secretion of VEGF and the metalloproteinase-9 (MMP-9), which are related to angiogenesis [13]. Therefore, high levels of TANs could provide a survival advantage for tumors, resulting in relapse and poor clinical outcomes for PC patients [105]. Blockade of CXCR2 in vivo hindered neutrophils’ entrance into PDAC stroma to significantly expand mouse survival [102]. Moreover, prevention of neutrophil maturation and migration by the tyrosine kinase inhibitor (TKI) lorlatinib was shown to abrogate PDAC development and metastasis in pre-clinical mouse models [106]. It was also shown that the high neutrophil/lymphocyte ratio (NLR) prior to therapy is associated with development of metastatic disease, and it could be used as a prognostic marker for OS in PC patients [107]. Additionally, neutrophils can form neutrophil extracellular traps (NETs). These structures consist of extracellular DNA released together with proteolytic enzymes that can enclose the tumor and inhibit the penetration of other anti-tumorigenic agents. NETs can also stimulate metastasis by attracting cancer cells from distant sites [13]. Treatment of PDAC mouse models with DNase I, a NET inhibitor, decreased the number of CAFs in the metastatic liver environment and thus suppressed metastasis [108]. Altogether, current evidence suggests that neutrophils can work synergistically with other cellular components to remodel the TME primarily in favor of pancreatic tumor growth. 

#### 3.3.5. Dendritic Cells

Dendritic cells (DCs) are trained antigen-presenting cells able to regulate anti-tumor immune responses by activating CD8^+^ and CD4^+^ T cells via MHC class I and II molecules, respectively [109]. DCs infiltrate pancreatic tumor lesions, and their abundance is associated with inhibition of disease progression [110]. The CD86 costimulatory marker expressed on DCs provides signals necessary for T cell activation and survival by binding to CD28 on the surfaces of T cells. However, cytotoxic T-lymphocyte-associated protein 4 (CTLA-4) produced by Tregs binds to CD86 with higher affinity than CD28, thus affecting CD8^+^ T cell activation and the recruitment of additional DCs [111]. The capacity of DCs is further impaired by PC cells through inhibition of their recruitment, maturation, and survival. The binding of CD154 (CD40L) on T helper cells to CD40 activates antigen-presenting cells. When DCs interact with cancer cells, immunosuppressive cytokines and chemokines, such as IL-10, TGF-β, and GM-CSF, are secreted, decreasing CD40 expression and keeping DCs in an immature state [112,113]. In contrary, pro-inflammatory cytokines and chemokines secreted by tumor cells, which are required for DCs activation, were downregulated by the activation of STAT3 in a melanoma mouse model [114]. A clinical study by Kobayashi et al. suggested that standard chemotherapy along with peptide-pulsed DC vaccines can act synergistically to improve PC patient survival [115].

#### 3.3.6. T Lymphocytes

T lymphocytes are mainly classified as CD8^+^ cytotoxic T cells (CTLs) and CD4^+^ helper T (Th) cells, which include Th1, Th2, Th17, and regulatory T cells (Tregs) [80]. Th1 cells promote cellular type I immunity against intracellular pathogens and tumors and Th2 cells are involved in the regulation of humoral type II immunity, whereas Th17 cells are the defense against extracellular pathogens. Finally, Tregs provide suppressive inflammatory responses to control autoimmunity, and they are able to diminish antitumor responses to promote tumor progression [13].

Pancreatic tumors are considered immunologically ‘cold’, displaying low infiltration of CD8^+^ CTLs that are localized along the invasive margin of the tumor border or in the surrounding fibrotic tissue [116]. PDAC patient tumors are usually abundant in Tregs that are inversely correlated with the presence of CD8^+^ CTLs and associated with poor clinical outcomes [117]. Furthermore, in the progression of different types of tumors, infiltrating CD8^+^ CTLs exhibit minimal activation and become exhausted. These non-functional CD8^+^ CTLs are characterized by impaired effector function, metabolism dysregulation, and less proliferative activity [118]. Additionally, TGF-β secretion in the TME inhibits CD8^+^ CTLs from producing cytolytic proteins, while PC cells often downregulate MHC-I expression, preventing recognition and cytotoxic activity by CD8^+^ CTLs [119]. 

In contrast to CD8^+^ CTLs, CD4^+^ helper T cells are a prominent feature of the infiltrated immune cells in the pancreatic TME. Th1 cells promote cell-mediated immune responses and are responsible for the activation of CD8^+^ T cells, NK cells, and M1-type macrophages [120]. On the other hand, Th2 lymphocytes assist humoral immune responses by producing a plethora of cytokines including IL-4, which contribute to the formation of the dense tumor stroma and the polarization of macrophages to M2 stage and promote PDAC progression [13]. The shift from Th1 to Th2 cells is a common characteristic in PDAC, correlated with decreased patient survival. Th2 skewing in PDAC is driven by the CAFs in stroma and cytokines such as IL-10 and TGF-β secreted by PC cells [117,119]. Furthermore, elevated numbers of Th17 cells in the tumor are associated with disease progression, and serum IL-17 levels are increased in PC patients and connected with disease severity [13,90]. Finally, Tregs are often found in higher numbers in pancreatic tumors, and usually contribute to an immunosuppressive environment that allows the tumor to evade the immune system and promote T cell exhaustion [89,121]. However, recent evidence in PDAC mouse models showed that Tregs depletion may not diminish immunosuppression but may promote tumor progression due to reduction in Tregs-mediated TGF-β secretion and subsequent loss of tumor-restraining fibroblasts. Moreover, upon Tregs reduction, chemokines CCL3, CCL6, and CCL8 were increased, resulting in restoration of immune responses [122,123]. Therefore, the role of Tregs in PC may be more complicated, and further studies are required in order to elucidate their detailed biological roles, depending on the cellular context. 

#### 3.3.7. B Lymphocytes

Through the expression of B cell receptors on their surface, B lymphocytes bind to foreign antigens and initiate an antibody response. Tumor-infiltrating B cells (TIL-Bs) complement T cell-mediated antitumor immunity [124]. In PDAC, elevated B cell infiltration is generally correlated with better prognosis, especially when those B cells cluster in tertiary lymphoid structures (TLS) [125]. However, the role of B lymphocytes in PDAC tumorigenesis remains controversial. This contradiction could be explained as B cells acquire different phenotypes during tumor progression. Their function is also determined by their localization in the TME, either scattered at the periphery of tumor or forming complexes with CD8^+^ T cells [126]. The immunosuppressive B cells (Bregs), which are responsible for restricting ongoing immune responses and reestablishing immune homeostasis, represent only a small fraction of the entire B cell population in PDAC [127,128]. TIL-Bs have been shown to be involved in PDAC initiation, progression, and fibrogenesis. B1 cells constitute a unique B cell subset with abnormal receptor signaling, an unusual resting location, stimulation of T cell expansion, induction of Th17 cell differentiation, and production of immunomodulatory IL-10 [129]. Upon pancreas-specific HIF1-α depletion, fibro-inflammatory stroma secretes CXCL13, leading to an influx of B1 regulatory B cells into the tumor and thus promoting carcinogenesis [87]. Treatment of HIF1-α-deficient mice with B cell-depleting αCD20 monoclonal antibodies prevented pancreatic intraepithelial neoplasia (PanIN) progression and development of invasive carcinomas [130]. Lastly, targeting Bruton tyrosine kinase (BTK) using the BTK inhibitor ibrutinib was shown to inhibit B cell- and M2 macrophage-mediated T cell suppression to decrease PC growth [131]. 

In summary, as discussed above, the highly desmoplastic nature of PC TME results in elevated mechanical forces which compress blood vessels, leading to hypoperfusion, hypoxia, and decreased infiltration of cytotoxic immune cells. Even the small number of anti-tumor immune cells present in TME is either exhausted or with an immature phenotype [132]. Immune evasion perpetrated by the tumor cells involves aberrant expression of immune and cancer cell surface markers, secretion of immunosuppressive cytokine and chemokine molecules in the TME, and activation of immune checkpoint pathways, as described below [133]. Therefore, considering the highly immunosuppressive pancreatic TME, new combinatorial therapeutic approaches are urgently needed to overcome PC immune tolerance (Figure 1). 

## 4. Current and Future Immunotherapy Strategies for Pancreatic Cancer

Cancer cells can survive and give rise to tumor development by, among other means, escaping immune surveillance either directly or indirectly via cells in the TME. Even though immunotherapy has revolutionized the treatment of various solid tumors during the last few years, it remains largely ineffective in PDAC patients by providing only a negligible improvement in patient survival [25,134,135]. The lack of PDAC responses to immunotherapies could be attributed, at least in part, to the low tumor mutation burden (TMB) in the vast majority of cases and to the highly desmoplastic TME, which collectively contribute to the development of an immunologically ‘cold’ environment [132]. PDAC has a highly desmoplastic TME with extensive fibrosis and extensive immunosuppression, which significantly compromises cytotoxic immune cell infiltration [58,136,137]. 

Under physiological conditions, immune system responses are regulated by various immune checkpoint pathways. Immune checkpoints are crucial modulators of the immune system, often exploited by cancer cells to evade immune surveillance. For example, programmed death protein–ligand 1 (PD-L1) or PD-L2 expressed by PC cells, MDCSs, or TAMs bind to PD-1 receptors on the surfaces of activated T cells, leading to T cell anergy or death [138,139]. Similarly, expression of CTLA-4 on the surface of T cells and binding to B7 molecules on DCs delivers an inhibitory signal that reduces T cell proliferation and activation and suppresses immune responses against tumor cells [140]. The upregulation of these inhibitory molecules and chronic antigen exposure leads to T cell exhaustion [13]. 

Over the last decade, several monoclonal antibodies targeting immune checkpoint molecules, such as PD-1 and CTLA-4, have been developed and granted FDA approval for the treatment of various solid tumors by reversing T cell dysfunction, leading to tumor killing [141,142]. Several studies have investigated the effects of anti-PD-1 (pembrolizumab, nivolumab), anti-PD-L1 (durvalumab), and anti-CTLA-4 (ipilimumab, tremelimumab) in PDAC as monotherapy or in combination with other approaches (Table 1). Favorable results regarding inhibition of tumor progression and improvement of patients’ survival observed in many cancer types were not observed in PDAC [134,135,143,144]. For the majority of PC patients, monotherapy treatment using PD-1 or CTLA-4 blockade has failed to produce any objective response, since some of the subjects experienced grade 3 or 4 adverse events related to treatment, and there were no responders [143,144]. The failure of ICB in pancreatic tumors is thought to be attributed, at least in part, to the low proportion of tumor-infiltrating T cells and the low tumor mutation burden (TMB) in PCs [73,145,146]. However, in a minor group of PDAC patients (~1%) with mismatch repair deficiency (dMMR) or microsatellite instability high (MSI-H), PD-1 blockade by pembrolizumab was shown to be effective and, currently, it is the only FDA-approved immunotherapy for patients with advanced PDAC. In this study, 8 out of the 86 patients had PC, and the objective response rate (ORR) among them was 62% (two patients had complete responses, three patients had partial responses, one patient had stable disease, and two patients were not evaluable) [147,148,149,150]. In the KEYNOTE-158 multi-cohort phase II study evaluating pembrolizumab, the median OS was 4 months in the PC subgroup, although the median duration of response was 13.4 months [151]. The next most promising outcomes of ICB were reported in PDAC patients after receiving anti-PD-1 together with gemcitabine/nab-paclitaxel, leading to PFS of 9.1 months and OS of 15 months [152]. Immunogenic cancer cell death includes the secretion of damage-associated molecular patterns from dying tumor cells that lead in the activation of tumor-specific immune responses, thus inducing long-term efficacy of anticancer drugs [153]. Hence, cytotoxic drugs may improve immunotherapeutic efficacy by stimulating immunogenic cancer cell death, decreasing tumor-induced immunosuppression, and enhancing effector T cell function and intra-tumoral infiltration [4]. 

A clinical trial (NCT01473940) examined the efficacy of ipilimumab combined with gemcitabine in patients with advanced PDAC, demonstrating a median PFS of 2.5 months and OS of 6.9 months, similar to gemcitabine treatment alone (6.8 months) [154]. Combinatorial treatment of tremelimumab with durvalumab or durvalumab monotherapy in PDAC patients yielded similar poor patient outcomes and no effect on disease progression [155]. Moreover, based on pre-clinical evidence showing the potential of Losartan as a stromal modifier able to reduce desmoplasia and enhance the intratumoral penetration and effectiveness of therapeutics in patients with PC [156], phase II clinical trials (NCT01821729, NCT05077800, and NCT03563248) are ongoing for the evaluation of losartan with ICB (nivolumab), chemotherapy (FOLFIRINOX), and radiotherapy in PC patients. Currently, there are various ongoing trials that explore the amalgamation of monoclonal antibodies with other therapies against PDAC (NCT05187338, NCT06353646, NCT05014776, NCT04117087, NCT05088889, NCT03816358, NCT03755739, NCT02834013) (Table 2).

Another immunotherapeutic strategy proposed in recent years for PC treatment is the use of vaccines. A specific anti-tumor immune response may be induced by presenting tumor antigens to the immune system in the form of a tumor-based vaccine. The vaccines clinically pursued in PDAC treatment mainly consist of whole-tumor cells, peptides, proteins, or recombinant constructs [13]. They may contain tumor-associated antigens (TAAs) or mutated tumor-specific antigens (TSAs) or neoantigens [157]. However, their mechanisms of action and efficacy depend on the TMB, which is limited in PDAC [158]. In a phase I trial, a variety of vaccines indicated no long-term survival benefit, although they broke tolerance and generated T cell immunity without any short-term adverse effects [159,160,161]. One of the best-studied therapeutic vaccines in PDAC is GVAX (granulocyte-macrophage colony-stimulating factor (GM-CSF) gene-modified tumor vaccine), which contains irradiated whole pancreatic tumor cells unable to grow that have been genetically modified to secrete GM-CSF. The results, so far, have demonstrated that GVAX is safe and able to induce antigen-specific T cell responses, with or without complementary treatments [4]. The preliminary results of NCT03161379 and NCT02648282 clinical trials showed that, in combination with immunochemoradiotherapy, GVAX is safe and can induce antigen-specific T cell responses [162]. However, GVAX has failed to provide long-term survival benefits and does result in improved treatment efficacy compared to standard-of-care chemotherapy [4]. Various agents, including viruses and bacteria, are currently being used to explore novel mechanisms that expose tumor antigens to the immune system using vaccines [163].

Furthermore, the use of cytokines and chemokines as immunomodulators is another approach in the immunotherapeutic armamentarium against cancer [164]. Agonistic CD40 therapy has been shown to polarize macrophages into a more tumoricidal M1 phenotype, leading to short-term survival benefits [165]. In a pre-clinical model of PDAC, a combination of agonist anti-CD40 with ICB improved survival by inducing T cell immunity and regression of subcutaneous tumors. This treatment combination almost doubled the survival of mice with spontaneous tumors, although they were not cured [13]. The most clinically important target of myeloid cells is C-C motif chemokine receptor 2 (CCR2), which controls the recruitment of inhibitory macrophages in the TME, and it is correlated with poor prognosis [81]. Mitchem et al. demonstrated that CCR2 blockade in PDAC enhances responses to chemotherapy, inhibits metastasis, and blocks monocyte access in the TME, which elevates the infiltration of T cells [166]. It was also shown that CD40 activation along with gemcitabine led to a partial response in PDAC patients, possibly by influencing the immune reaction via TAMs [165]. In a phase I trial, the CCR2 inhibitor PF-04136309 indicated a clinical response in half of PDAC patients pre-treated with FOLFIRINOX [167]. In addition, the pre-clinical analysis of colony-stimulating factor 1 receptor (CSF-1R), a key cytokine regulator of MDSCs and TAMs in PDAC, in combination with checkpoint inhibition using PD1 and CTLA-4, displayed favorable results [168]. Lastly, after CXCR2 inhibition, which is responsible for neutrophil and MDSC migration, T cell infiltration was enhanced. When the CXCR2 blockade was combined with either CSF-1R inhibition or checkpoint blockade, tumor responses were improved [169,170]. Moreover, a dual inhibitor targeting both the Bromo- and extra-terminal Motif (BET) protein BRD4 and histone acetyltransferase EP300/CBP was found to inhibit oncogenic Ras signaling and enhance the efficacy of anti-PD-1 ICB in PC mouse models [171]. Interestingly, a recent study showed that combining anti-4-1BB and anti-LAG3 ICBs with a CXCR1/2 inhibitor targeting myeloid cells can overcome immunotherapy resistance and result in durable therapeutic responses in genetically engineered and syngeneic mouse models for PDAC [172]. On the other hand, in a cohort study of 69 patients with resected PDAC, it was demonstrated that although T cells were associated with prolonged DFS, lymphocyte-activation gene 3 (LAG-3) expression by PDAC-infiltrating T cells was correlated with reduced DFS [173]. 

Adoptive cell therapies constitute an alternative approach against pancreatic tumors by using autologous or allogeneic immune effector cells, including T cells, DCs, or NKs, to eradicate cancer cells [4]. After being isolated from the patient’s blood or a tumor, these immune cells are genetically modified ex vivo to stimulate an anti-tumor immune response and injected back into the patient [4,30]. Expanding tumor-infiltrating T cells that can migrate from the vasculature into the TME may be beneficial [174]. Immune cells could also be engineered in vitro to replace non-functional lymphocytes. Specifically, chimeric antigen receptor (CAR)-T cells select a cancer cell-specific surface protein as a target to generate effective therapy by removing T cells and reducing competition for stimulatory cytokines [175]. CAR-T cells are typically infused systemically to target malignant cells and exert anti-tumor activity [176]. Several trials for treating solid tumors including PDAC, such as NCT02850536 and NCT02349724, using CAR-T cells have been completed, with preliminary results exhibiting the safety of regionally infused CAR-T cells, but limited biological efficacy [177]. A pre-clinical study in a murine PDAC model demonstrated promising data after treatment with mesothelin-directed CAR-T cells together with oncolytic viruses expressing IL-2 and tumor necrosis factor-α (TNF-α) [178]. Clinically, CAR-T cell therapy has not yet been extensively explored in PDAC. Although CAR-engineered cells show early promise in hematological malignancies, their efficacy in solid tumors is still limited, but is under active investigation [176]. Importantly, this approach is not risk-free, as it may cause immune hyperstimulation, which can be fatal in some cases [179]. A multifaceted approach involving this cellular monotherapy is required in order to identify the appropriate personalized treatment for each PDAC patient.

**Table 1 ijms-25-09555-t001:** Summary of completed clinical trials in pancreatic cancer immunotherapy.

Drug(s)	Mechanism	Population	Clinical Trial/Year of Completion	Phase of Trial	Trial Design/Number of Patients	Primary Endpoint
CP-870,893 in combination with gemcitabine	CD40 agonist antibody with chemotherapy	Chemotherapy-naïve, surgically incurable PC	NCT00711191 01/2011	I	Single group assignment/22	Dose-limiting toxicities (DLTs), adverse events [165]
Ipilimumab (BMS-734016) and PC vaccine	CTLA-4 inhibitor with allogeneic pancreatic tumor cells transfected with a GM-CSF gene	Locally advanced, unresectable, or metastatic pancreatic adenocarcinoma	NCT00836407 07/2012	I	Two-arm trial: Ibilimumab alone and Ibilimumab plus PC vaccine/30	Unacceptable toxicity [180]
MEDI4736 in combination with nab-paclitaxel and gemcitabine or with AZD5069	Anti-PD-L1 monoclonal antibody with chemotherapy or CXCR2 antagonist	Metastatic PDAC	NCT02583477 07/2018	I/II	Parallel assignment/23	Adverse events, DLTs, ORR
TG01/GM-CSF with gemcitabine	TG01 and granulocyte macrophage colony-stimulating factor with chemotherapy	Resected adenocarcinoma of the pancreas	NCT02261714 05/2019	I/II	Single group assignment/32	Immune responses, adverse events [181]
Ipilimumab, vaccine, FOLFIRINOX	CTLA-4 inhibitor with allogeneic GM-CSF-transfected pancreatic tumor vaccine against chemotherapy	Metastatic PC	NCT01896869 05/2019	II	Two-arm trial: Ibilimumab plus vaccine or FOLFIRINOX alone/83	Overall survival (OS)
FOLFIRINOX, Losartan, Proton Beam Radiation	Combination of 5-fluorouracil, leucovorin and oxaliplatin with proton beam therapy	Locally advanced disease	NCT01821729 09/2021	II	Single group assignment/50	Number of participants with R0 resection
Cyclophosphamide, GVAX PC vaccine, Pembrolizumab, radiation	Chemotherapy, GM-CSF-secreting allogeneic PC vaccine with PD-1 blockade antibody and stereotactic body radiation therapy	Locally advanced adenocarcinoma of the pancreas	NCT02648282 01/2022	II	Single group assignment/58	Distant metastasis-free survival
Oleclumab (MEDI9447), Durvalumab, Gemcitabine, Nab-paclitaxel, Oxaliplatin, Folinic acid, 5-FU	Anti-CD73 monoclonal antibody, anti-PD-L1 monoclonal antibody in combination with chemotherapy	Metastatic PDAC	NCT03611556 07/2022	I/II	Parallel assignment/213	TEAEs, TESAEs, DLTs, abnormal vital signs, electrocardiogram and clinical laboratory parameters, ORR
Plerixafor and Cemiplimab	CXCR4 inhibitor (small molecule) and PD-1 blocking antibody	Metastatic PC	NCT04177810 05/2023	II	Single group assignment/25	ORR
Cyclophosphamide, Nivolumab, Ipilimumab, GVAX Pancreas Vaccine, CRS-207	Chemotherapy plus immunotherapy (anti-PD-1 and anti-CTLA-4) plus PC vaccine	Previously treated metastatic pancreatic adenocarcinoma	NCT03190265 08/2023	II	Parallel assignment with or without PC vaccine/61	Objective response rate (ORR)

**Table 2 ijms-25-09555-t002:** Summary of ongoing clinical trials in pancreatic cancer immunotherapy.

Drug(s)	Mechanism	Type of Disease	ID	Phase	Trial Design/Number of Patients	Primary Endpoint
Ipilimumab in combination with pembrolizumab and durvalumab	Combination of three antibodies against PD1, PDL1, and CTLA4	Advanced solid tumors	NCT05187338	I/II	Single group assignment/100 (estimated)	Drug safety, progression-free survival (PFS), disease control rate, duration of remission
XH001 combination with Ipilimumab and gemcitabine /capecitabine	mRNA neoantigen cancer vaccine in combination with immunotherapy that targets CTLA-4 protein on T cells plus chemotherapy	Resected PC	NCT06353646	N/A	Single-center, open-label, single-arm/12 (estimated)	Efficacy and safety trial
Tadalafil, Pembrolizumab, Ipilimumab, CRS-207	PD-1/CTLA-4-blocking antibodies in combination with immunotherapy	Previously treated metastatic PDAC	NCT05014776	II	Single group assignment/17 (actual)	Objective response rate (irORR) using immune response evaluation criteria for solid tumors (iRECIST)
KRAS peptide vaccine Nivolumab Ipilimumab	KRAS peptide vaccine with poly-ICLC adjuvant in combination with immunotherapy	Resected mismatch repair protein (MMR-p), Colorectal and PC	NCT04117087	I	Parallel assignment/30 (estimated)	Number of participants experiencing study drug-related toxicities Fold change in interferon-producing mutant-KRAS-specific cytotoxic (CD8) and helper (CD4) T cells at 16 weeks
Nivolumab Ipilimumab Stereotactic body radiation therapy Low-dose irradiation	Cytotoxic chemotherapy followed by hypofractionated radiotherapy to sensitize pancreatic cancer to immunotherapy consisting of combined PD-1 and CTLA4 blockade	First-line treatment stage IV pancreatic cancer	NCT05088889	I	Study arm/10 (estimated)	Objective tumor response rate 1 (ORR1) in study patients, assessed by RECIST v1.1 Objective tumor response rate 2 (ORR2) after first progression, assessed by RECIST v1.1
Combination of anetumab ravtansine with Either nivolumab, nivolumab and ipilimumab, or gemcitabine and nivolumab	Monoclonal antibody linked to a chemotherapy drug called DM4 attaches to mesothelin-positive cancer cells and delivers DM4 to kill them. Immunotherapy chemotherapy	Advanced PC	NCT03816358	I	Parallel assignment/ 74 (estimated)	Maximum tolerated dose (MTD)
Checkpoint inhibitor (CPI) such as Pembrolizumab plus chemotherapy	Trans-artery/intra-tumor infusion of PD1/PDL1 antibody and/or CTLA4 antibody ipilimumab plus chemotherapeutic drug and comparison of their differences.	Advanced solid tumors	NCT03755739	II/III	Parallel assignment/200 (estimated)	Overall survival Complete response (CR) rate before or at month 6
Nivolumab and Ipilimumab	Immunotherapy with monoclonal antibodies	Rare pancreatic tumors including acinar cell carcinoma, mucinous cystadenocarcinoma, or serous cystadenocarcinoma	NCT02834013	II	Parallel assignment/818 (estimated)	ORR
9-ING-41 Losartan Ferumoxytol FOLFIRINOX	Blocking of GSK-3β activity using 9-ING-41 and blocking of TGF-β function using Losartan to inhibit cancer cell resistance to FOLFIRINOX chemotherapy	Metastatic PDAC without prior therapy	NCT05077800	II	Single group assignment/70 (estimated)	PFS
Losartan and nivolumab in Combination with FOLFIRINOX and SBRT	Combination chemotherapy. Losartan is used to lower blood pressure. Nivolumab is an antibody that may cause apoptosis. Radiation by stereotactic body radiation therapy	Localized PDAC; borderline/potentially resectable or locally advanced.	NCT03563248	II	Parallel assignment/168 (actual)	Proportion of participants with R0 resection
Motixafortide Cemiplimab Gemcitabine Nab-paclitaxel	Combination chemotherapy (gemcitabine and nab-paclitaxel), chemokine (C-X-C) motif receptor 4 inhibitor (BL-8040), and immune checkpoint blockade (Cemiplimab)	Metastatic treatment-naïve PDAC	NCT04543071	II	Single group assignment/10 (estimated)	Overall response rate (complete response (CR) + partial response (PR))
Durvalumab Rintatolimod	Combination therapy of ICI therapy with a toll-like receptor 3 (TLR-3) agonist	Metastatic PC	NCT05927142	I/II	Sequential assignment/43 (estimated)	Phase Ib: Determine safety of combination therapy with durvalumab and rintatolimod Phase II: Determine the clinical benefit rate of combination therapy with durvalumab and rintatolimod.
PT199 Tislelizumab	PT199 counters the adenosine-mediated immunosuppressive TME, rendering anti-tumor immune cells more responsive to checkpoint immunotherapies, such as PD-1/PD-L1 inhibitors	Advanced solid tumors	NCT05431270	I	Sequential assignment/40 (estimated)	MTD, DLT, Safety of PT199
IMM-101, Pembrolizumab, Gemcitabine	Combination of a heat-inactivated mycobacterium, immune modulator with chemotherapy	Metastatic PC	NCT06498518	II	Single group assignment/50 (estimated)	ORR
Ipilimumab Nivolumab Radiation Therapy	Ipilimumab inhibits cancer cell growth. Nivolumab induces apoptosis. Radiation therapy may increase the likelihood of response to interventions.	Metastatic, microsatellite-stable PC	NCT04361162	II	Single group assignment/30 (estimated)	ORR
Pembrolizumab With Olaparib	PD-1 inhibitor in combination with PARP inhibitor	Metastatic PDAC with mismatch repair deficiency or tumor mutation burden > 4 mutations/Mb	NCT05093231	II	Single group assignment/20 (estimated)	ORR
Nivolumab Ipilimumab SBRT TGFβ-B-15 peptide	Combination of checkpoint-blocking antibodies with immunomodulation of the TME; TGFβ-15 immune response is correlated to clinical benefit	Refractory PC	NCT05721846	I	Single group assignment/20 (estimated)	Adverse effects
Pembrolizumab Folfirinox SABR	Combination of chemotherapy, PD-1 inhibition, and radiotherapy	Borderline resectable PC	NCT06384560	I/II	Single group assignment/66 (estimated)	Percentage of patients with progression free survival at 18 months (RECIST 1.1)
Lenvatinib Pembrolizumab	Inhibition of cell growth, PD-1 inhibition	Advanced unresectable PC	NCT04887805	II	Single group assignment/28 (estimated)	PFS
Pembrolizumab With Olaparib/Olaparib alone	PD-1 inhibitor in combination with PARP inhibitor	Advanced PC with germline BRCA1 or BRCA2 mutations	NCT04548752	II	Parallel assignment /88 (estimated)	PFS
Pembrolizumab Defactinib	Reprogramming the TME by targeting FAK following chemotherapy to potentiate anti-PD-1 antibody	Resectable PDAC	NCT03727880	II	Parallel assignment /36 (estimated)	Pathologic complete response (pCR) rate
SBRT Nivolumab CCR2/CCR5 dual antagonist GVAX	Combination therapy to enhance the infiltration of CD8+CD137+ cells in PDAC	Locally advanced PDAC	NCT03767582	I/II	Sequential assignment /30 (estimated)	Number of Participants experiencing study drug-related toxicities Percentage of participants treated with immunotherapy who achieve an immune response
Nivolumab Ipilimumab SBRT Low dose irradiation	Cytotoxic chemotherapy followed by hypofractionated radiotherapy to sensitize PC to combined PD-1 and CTLA4 blockade	Stage IV PDAC	NCT05088889	I	Single group assignment/10 (estimated)	Objective tumor response rate 1 (ORR1) in study patients, assessed by RECIST v1.1 Objective tumor response rate 2 (ORR2) after first progression, assessed by RECIST v1.1
Nivolumab Ipilimumab Hydroxychloroquine (HCQ)	Immunotherapy in combination with standard-of-care chemotherapy	Previously untreated PDAC	NCT04787991	I	Parallel assignment/45 (estimated)	Incidence and severity of adverse events
Tumor-Infiltrating Lymphocytes Pembrolizumab	Autologous tumor-infiltrating lymphocytes following a lymphodepleting regimen plus PD-1 inhibition	Metastatic PC	NCT01174121	II	Parallel assignment/332 (estimated)	Response rate
Epacadostat Pembrolizumab CRS-207	IDO-1 inhibition, PD-1 inhibition, stimulation of immune response to mesothelin	Metastatic PC	NCT03006302	II	Parallel assignment/ 41 (estimated)	MTD 6-month survival
Atezolizumab Tivozanib	Inhibition of VEGF, inhibition of PD-1	Immunogenically cold PDAC	NCT05000294	I/II	Sequential assignment/29 (estimated)	ORR
Gemcitabine, Nab-paclitaxel, Capecitabine, Cisplatin, Irinotecan Olaparib and Pembrolizumab	Low-dose chemotherapy followed by maintenance with PD-1 and PARP inhibition	Metastatic untreated PDAC	NCT04753879	II	Single group assignment/38 (estimated)	PFS
mFOLFIRINOX Verteporfin Pembrolizumab	Photoradiation with verteporfin, PD-1 inhibition plus standard-of-care chemotherapy	Locally Advanced or Metastatic PC	NCT06381154	II	Single group assignment /25 (estimated)	ORR
Nivolumab Ipilimumab Radiation Therapy	Blocking PD-1/PD-L1 and CTLA-4 pathways	PC	NCT03104439	II	Single group assignment /80 (estimated)	Disease control rate
Gemcitabine Nab-paclitaxel Nivolumab	ICIs with chemotherapy	Borderline resectable, locally advanced or mPC	NCT03970252	I/II	Single group assignment /40 (estimated)	Incidence of treatment-related AEs, SAEs, AEs leading to discontinuation, death, and laboratory abnormalities
Futibatinib Pembrolizumab Cisplatin	ICIs with chemotherapy	PDAC	NCT05945823	II	Parallel group assignment /40 (estimated)	ORR
LGK974 PDR001	PORCN inhibitor PD-1 inhibitor	PC that has progressed despite standard therapy or for which no effective standard therapy exists	NCT01351103	I	Single group assignment/185 (actual)	Maximum tolerated dose or recommended dose for expansion of LGK974 as a single agent or in combination with PDR001 in treated patients

## 5. Conclusions

Pancreatic cancer remains one of the deadliest diseases, with a dismal prognosis. Currently, prevention or early diagnosis at a curable stage is extremely difficult. Failure of approved PDAC therapies has dramatic effects on the quality of life of cancer patients. One of the emerging interests in PDAC therapy is immunotherapy due to the capability of cancer cells to escape immune surveillance through various mechanisms, including their densely packed TME that is depleted of effector T cells. Numerous evidence suggests the importance of unraveling the complexity of TME components and their multifaceted interactions in tumor suppression or progression. Findings from clinical trials encompassing immunotherapeutic strategies have not been encouraging regarding a cure for PDAC. Moreover, some treatments have shown partial effectiveness against PDAC, such as FOLFIRINOX, but most of them have failed to provide a significant survival improvement without serious side effects. Hence, future treatment approaches should aim to comprise therapies that target multiple characteristics of the TME simultaneously. To this end, ongoing efforts to evaluate the efficacy of remodeling the pancreatic TME using multimodal strategies might be proven beneficial for a subset of PDAC patients to provide an increased survival rate and/or better quality of life. However, an increased number of more sizeable studies are needed to clarify the effects of different treatment options and the optimal therapeutic protocols to combine them. Various trials have failed, but the progress in our knowledge about pancreatic TME gives substantial hope for the future development of successful therapies. An improved understanding of the crosstalk between tumor, stromal, and immune cells may soon lead to discoveries able to reverse the innate resistance of PDAC to immunotherapies.

## Figures and Tables

**Figure 1 ijms-25-09555-f001:**
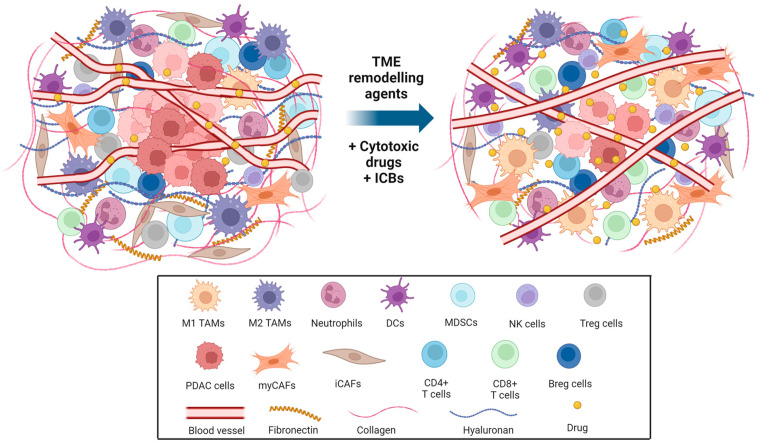
Pancreatic tumor microenvironment remodeling strategies to improve therapeutic efficacy. Uncontrolled proliferation of pancreatic cancer cells within desmoplastic stroma, established by excessive deposition of ECM components such as collagen, hyaluronan, and fibronectin, results in the accumulation of mechanical forces and collapsed blood vessels. As a result, abnormal vascularization and ECM stiffness impair vessel perfusion, tumor oxygenation, and drug delivery. In addition, this hinders the infiltration of cytotoxic immune cells, such as CD8+ T and NK cells, and creates a highly immunosuppressive TME along with the secretion of inflammatory cytokines and chemokines by iCAFs, Tregs cells, MDSCs, Bregs, and M2-type TAMs. ECM remodeling strategies using anti-fibrotic drugs and reprogramming of the immunosuppressive TME using immunomodulatory agents could be used synergistically to reverse this phenomenon. This combinatorial approach could normalize tumor vasculature and enhance vessel perfusion and oxygenation, followed by increased infiltration of CD8+ T and NK cells and a decrease in immunosuppressive Tregs, MDCSs, and iCAFs, as well as polarization of TAMs towards an anti-tumor M1 phenotype. Collectively, these strategies aim to enhance pancreatic anti-tumor immunity and the efficacy of ICBs in combination with chemo- and/or nanotherapy to significantly improve pancreatic cancer immunotherapy and patient outcomes. Created with BioRender.com. TME: tumor microenvironment; ICBs: immune checkpoint blockers; M1 TAMs: M1 type tumor-associated macrophages; M2 TAMs: M2 type tumor-associated macrophages DCs: dendritic cell; MDSCs: myeloid-derived suppressor cells; NK cells: natural killer cells; Treg cells: regulatory T cells; PDAC cells: pancreatic ductal adenocarcinoma cells; myCAFs: myofibroblastic cancer-associated fibroblasts; iCAFs: inflammatory cancer-associated fibroblasts; CD4+ T cells: CD4+ cytotoxic T cells, CD8+ T cells: CD8+ cytotoxic T cells; Breg cells: regulatory B cells.

## Data Availability

No new data were created as part of this review paper.

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
