# Peer review of "The Role of Tumor Microenvironment in Pancreatic Cancer Immunotherapy: Current Status and Future Perspectives"

_ijms, 2024, doi:10.3390/ijms25179555_

Round 1

Reviewer 1 Report

Comments and Suggestions for Authors

This is a well-written and comprehensive review of pancreatic cancer (PC). The amount of effort that has been put in to put this review together is commendable.

However, I would like to provide a few suggestions that may enhance the manuscript further:

  1. While the title of the manuscript is intriguing and focused, the content of the review seems somewhat broad and general. For instance, the detailed discussion of various standard-of-care (SOC) treatments for PC might extend beyond the intended scope of this review.
  2. I notice that some parts of the manuscript share similarities with some existing reviews (like PMID: 34638513). It might be beneficial to incorporate more of the authors' unique perspectives to distinguish this review from others.
  3. Consider adding the publication years to Table 1. Given the large number of entries and the lack of a specific order, including the years could help readers navigate the information more effectively.

Author Response

Reviewer comments 1:

This is a well-written and comprehensive review of pancreatic cancer (PC). The amount of effort that has been put in to put this review together is commendable.

However, I would like to provide a few suggestions that may enhance the manuscript further:

  1. While the title of the manuscript is intriguing and focused, the content of the review seems somewhat broad and general. For instance, the detailed discussion of various standard-of-care (SOC) treatments for PC might extend beyond the intended scope of this review.
  2. I notice that some parts of the manuscript share similarities with some existing reviews (like PMID: 34638513). It might be beneficial to incorporate more of the authors' unique perspectives to distinguish this review from others.
  3. Consider adding the publication years to Table 1. Given the large number of entries and the lack of a specific order, including the years could help readers navigate the information more effectively.

Response 1:

We would like to thank the reviewer for the positive comments and constructive feedback.

  1. Thank you for the useful comment. We understand that some parts of the review, including various standard-of-care treatments were too detailed and lengthy, considering that they are not within the direct scope of the review. To address this, we have now shortened this part, mainly by removing excessive information regarding surgery and chemotherapy (lines 74-124, see the manuscript with tracked changes). However, we believe that the information regarding established guidelines and treatment protocols along with the extent of their effectiveness, is important to set the current therapeutic framework and highlight the urgent need for improving pancreatic cancer immunotherapy in combination with other approved treatments, as it was also recommended by other reviewers.
  2. Thank you for pointing this out. While our review paper shares some conceptual similarities with some other recently published papers, we believe that it provides a unique combination of information on this emerging topic. It initially provides the current framework and guidelines for PDAC treatment. Moreover, it not only discusses the latest information regarding the immunomodulatory interactions between cancer and various immune cell components but also focuses on the importance of PDAC mechanobiology and ECM constituents in desmoplasia, hypoperfusion, and solid stress leading to the development of an immunosuppressive TME which limit the efficacy of immunotherapy. Finally, it provides a comprehensive description of the completed as well as the most current ongoing clinical trials aiming to test different combinatorial immunotherapy treatment protocols based on the unique features of pancreatic tumors. Therefore, we are confident that it would be of great interest to researchers and clinicians working in this rapidly evolving field.
  3. Thank you for this suggestion. We have now included the “Year of completion” for every clinical trial in Table 1 and listed all completed trials from the oldest to the latest, to facilitate readers navigating through this information.

Reviewer 2 Report

Comments and Suggestions for Authors

Pancreatic cancer (PDAC) is one of the deadliest cancers in the world. In this manuscript, the authors reviewed the current therapies for pancreatic cancer, pancreatic tumor microenvironment and opportunities for therapeutic interventions, and current and future immunotherapy strategies for pancreatic pancer. Overall, this is a relatively completed summary and the manuscript is well-written and I have no specific comments.

Author Response

Comments 2:

Pancreatic cancer (PDAC) is one of the deadliest cancers in the world. In this manuscript, the authors reviewed the current therapies for pancreatic cancer, pancreatic tumor microenvironment and opportunities for therapeutic interventions, and current and future immunotherapy strategies for pancreatic cancer. Overall, this is a relatively completed summary and the manuscript is well-written and I have no specific comments.

Response 2:

We would like to thank the reviewer for the positive comments.

Reviewer 3 Report

Comments and Suggestions for Authors

The authors presented the review entitled” The role of tumor microenvironment in pancreatic cancer immunotherapy: Current status and future perspectives”.

I have suggestions.

In introduction section, guideline of pancreatic cancer treatment should be briefly described.

Figure 1. Pancreatic tumor microenvironment remodeling strategies to improve therapeutic efficacy. If the text related to the figure (legends), (Figure) should be added in the text. Abbreviations should be described in the figure legend.

Comments on the Quality of English Language

Minor editing of English language required.

Author Response

Comments 3:

The authors presented the review entitled” The role of tumor microenvironment in pancreatic cancer immunotherapy: Current status and future perspectives”.

I have suggestions.

In introduction section, guideline of pancreatic cancer treatment should be briefly described.

Figure 1. Pancreatic tumor microenvironment remodeling strategies to improve therapeutic efficacy. If the text related to the figure (legends), (Figure) should be added in the text. Abbreviations should be described in the figure legend.

Response 3:

We would like to thank the reviewer for the positive comments and constructive feedback.

  1. According to the suggestion, we have now included a short section at the end of the introduction, briefly describing a summary of the current guidelines for pancreatic cancer treatment (lines 52-61, see attached manuscript with tracked changes).
  2. We have now referred to Figure 1 in the text (line 454) and have described all abbreviations at the end of Figure 1 legend (lines 610-616, see the attached manuscript with tracked changes).

Reviewer 4 Report

Comments and Suggestions for Authors

In this review paper, Poyia et al. provided a comprehensive overview of the intricate relationship between pancreatic ductal adenocarcinoma (PDAC) and the tumor microenvironment (TME). The authors summarized the connection between the highly immunosuppressive nature of the TME, the poor prognosis of PDAC, and the limited success of current therapies. The review comprehensively highlighted various components of the TME and their roles in supporting tumor progression and resistance to therapy. The authors also discussed ongoing and future therapeutic strategies and clinical trials to enhance the efficacy of immunotherapy, including antibodies, vaccines, immunomodulators, and CAR-T cells.

This review is of high quality and thoroughly covers the biological and clinical aspects of PDAC. These detailed analyses are crucial for understanding the complexity of PDAC. The authors focused on both the challenges posed by the TME and the potential therapeutic avenues, especially immunotherapy, which makes it a valuable resource for researchers and clinicians interested in pancreatic cancer. I recommend acceptance of this comprehensive review.

Author Response

Comments 4:

In this review paper, Poyia et al. provided a comprehensive overview of the intricate relationship between pancreatic ductal adenocarcinoma (PDAC) and the tumor microenvironment (TME). The authors summarized the connection between the highly immunosuppressive nature of the TME, the poor prognosis of PDAC, and the limited success of current therapies. The review comprehensively highlighted various components of the TME and their roles in supporting tumor progression and resistance to therapy. The authors also discussed ongoing and future therapeutic strategies and clinical trials to enhance the efficacy of immunotherapy, including antibodies, vaccines, immunomodulators, and CAR-T cells.

This review is of high quality and thoroughly covers the biological and clinical aspects of PDAC. These detailed analyses are crucial for understanding the complexity of PDAC. The authors focused on both the challenges posed by the TME and the potential therapeutic avenues, especially immunotherapy, which makes it a valuable resource for researchers and clinicians interested in pancreatic cancer. I recommend acceptance of this comprehensive review.

Response 4:

We would like to thank the reviewer for the enthusiastic comments, they are well-appreciated.